# Ecotoxicological Effects of Pesticides on Hematological Parameters and Oxidative Enzymes in Freshwater Catfish, *Mystus keletius*

**Ayyanar Barathinivas [1], Subramanian Ramya [1], Kooturan Neethirajan [1], Ramaraj Jayakumararaj [2], Chinnathambi Pothiraj [2], Paulraj Balaji [3] and Caterina Faggio [4,\*]**

[1] PG and Research Department of Zoology, Yadava College (Men), Madurai 625 014, Tamil Nadu, India; nivibha@gmail.com (A.B.); sjramya84@gmail.com (S.R.); neethirajank56@yahoo.com (K.N.)

[2] Department of Botany, Government Arts College, Melur, Madurai 625 106, Tamil Nadu, India; jayakumar74@gmail.com (R.J.); pothirajchinnathambi71@gmail.com (C.P.)

[3] PG and Research Centre in Biotechnology, MGR College, Hosur 635 130, Tamil Nadu, India; balaji_paulraj@yahoo.com

[4] Department of Chemical, Biological, Pharmaceutical and Environmental Sciences, University of Messina, Viale Ferdinando Stagno d'Alcontres, 31, 98166 Messina, Italy

\* Correspondence: cfaggio@unime.it

**Abstract:** Hematological parameters and changes in stress-induced functionalities of cellular enzymes have been recognized as valuable tools for monitoring fish health and determining the toxic effects of pesticides. The present study was conducted to evaluate the toxic effect of selected pesticides viz., Ekalux (EC-25%), Impala (EC-55%), and Neemstar (EC-15%) on freshwater catfish *Mystus keletius*. Fish were exposed to sub-lethal concentrations (mg/L) of the selected pesticide for a period of 7, 14, 21, and 28 days. Hematological parameters viz., total erythrocyte (RBC), hemoglobin (Hb), and hematocrit (Ht) packed cell volume values decreased with an increase in exposure time to pesticides, whereas the values for parameters viz., leucocytes (WBC), mean corpuscular hemoglobin concentration (MCHC), mean corpuscular volume (MCV), and mean corpuscular hemoglobin (MCH) increased significantly. A decrease in packed cell volume (PCV) and hemoglobin values coupled with decreased and deformed erythrocytes as signs of anemia were also observed. The effect of pesticides on RBC content was 1.43 (million/mm$^3$) on day 7 and reduced to 1.18 (million/mm$^3$) on days 14 and 21. A similar trend was found for Impala on RBC, which had an initial value of 1.36 (million/mm$^3$) on day 7 and reached a value of 1.10 (million/mm$^3$) on day 28. In contrast, the value of Neemstar decreased from 1.59 (million/mm$^3$) on day 7 in control to 1.02 (million/mm$^3$) on day 28. Data indicates that the order of toxic effect of pesticides recorded a maximum for Impala followed by Ekalux and Neemstar in the selected fish model. Likewise, the overall pattern of pesticidal activity on cellular enzymes (GDH, MDH, and SDH) recorded a maximum toxic effect for Impala followed by Ekalux and Neemstar. Results indicate that Chlorpyrifos pesticide-Impala evoked maximum toxic effect on selected tissues compared to the other two pesticides tested. Statistical analysis of the summative data using two way ANOVA was statistically significant (*p*-value < 0.001). The differences in the hematological parameters analyzed are attributed to the physiological acclimatization of the fish to the local conditions, which influences the energy metabolism and consequently determines the health status of the fish. Overall, Impala exhibited the highest pesticidal activity on cellular enzyme, followed by Ekalux and Neemstar. Results suggest that natural pesticides may be preferable for rice field application in terms of environmental safety.

**Keywords:** ecotoxicology; blood parameters; respiratory enzymes; Ekalux; Impala; Neemstar; *Mystus keletius*

## 1. Introduction

The indiscriminate application of synthetic chemical pesticides in agricultural land has detrimental impacts on the water's odor and taste, as well as a lethal impact on non-target creatures in the aquatic system [1–4]. Freshwater ecosystem contamination is among the most serious threats to aquatic living organisms, as water sources serve as the intended destination for a complex interplay of agrochemicals and other xenobiotics [4–8]. In India, the expansion of the agro-industrial sector coupled with an increase in the application of agrochemicals (herbicides, insecticides, pesticides, fungicides) has increased over a period of time to meet the growing demand for agro-industrial products [9,10]. Organophosphorus (OP) pesticides are favored at the moment and have largely supplanted organochloro insecticides. Glyphosate is the most widely used herbicide for farming applications in southern India, whereas chlorpyrifos [O,O-diethyl O-(3,5,6-trichloro-2-pyridinyl) phosphorothioate] is the most widely used insecticide. Although there are numerous eco-friendly biopesticides, integrated pest management systems, neem-based biopesticides, and other natural pesticides available for IPM, farmers continue to rely on chemical pesticides due to their immediate effect and wide availability [11–15]. Once applied, a significant portion of pesticides in rice fields might reach surrounding aquatic areas, where they can have a detrimental effect on the biota. These contaminants can result in environmental stress, which frequently results in inappropriate regulatory responses in aquatic creatures, particularly edible fish [16–21].

Blood assessment is an essential technique for determining the physiological health of fish. Hematological measurements can be used to detect changes in a characteristic that exceeds its usual homeostatic limitations [22]. The erythrocyte count (RBC), hemoglobin concentration (Hb), hematocrit value (Ht), mean corpuscular volume (MCV), mean corpuscular hemoglobin (MCH), and mean corpuscular hemoglobin level (MCHC) are all red blood cell parameters. Leukocyte count (WBC) and, on occasion, differential leukocyte count (DLC) are used to determine the proportion or number of several types of leukocytes: lymphocytes (Lym), neutrophils or heterophils (Neu), monocytes (Mono), eosinophils (Eos), and basophils (Bas) (Bas). Bioaccumulation and oxidative stress produced by pesticides in fish are also discussed [23–27]. Among the many marker enzymes, catalase (CAT) and superoxide dismutase (SOD) are critical indicators of pesticide-induced oxidative stress. As a result, oxidative enzymes are frequently utilized as biological markers to investigate the effect of harmful materials on aquatic species, such as pesticides [27–30]. Stress can alter the values of several red blood cell parameters (Ht, Hb, RBC, and MCV) as well as a variety of biochemical markers (e.g., glucose, catecholamine, and cortisol levels). Increased values of red blood parameters frequently occur as a result of stress, as stress reactions need increased energy expenditure ("fight or flight"), and increased oxygen transport is one of the adaptive processes associated with stress [22]. Hematological and biochemical indicators offer precise data about the oxygen transport capacity of fish, their immunological potential, their stress level, sickness, intoxication, and nutritional status, among other things. In fish, red blood cell properties can be altered by erythrocyte swelling (increased MCV and Ht), splenic erythrocyte discharge, or, over a longer period of time, increased erythropoiesis (increase in Ht, Hb, and RBC). Dobsikova et al. [31], Fazio et al. [32], and Aguirre-Guzman et al. [33] all observed an increase in red blood cell indices in response to stress. Similarly, synthetic pesticides have a sublethal effect on the metabolic pathways and protein composition of non-target taxa. Ecotoxicological indicators are extensively utilized in toxicology bioassays, as well as in environmental stewardship and threat assessment methods, as early-warning indicators of contaminants' detrimental sublethal effects. Additionally, the amount of change can be used to measure the amount of anxiety [34].

## 2. Materials and Methods

### 2.1. Pesticides and Chemicals Used

Purchased from local farm supply stores, the pesticides Ekalux, Impala, and Neemstar were utilized in the study. All the standards used in the study were purchased from Sigma

Chemical Co. (St. Louis, MO, USA), and all other chemicals used were purchased from Merck. All the chemicals and reagents used in the study were analytical grade.

*2.2. Pesticides Used*

2.2.1. Ekalux–Quinalphos

Ekalux EC-25 (Pesticide Corporation of India Ltd., Uttar Pradesh, India) is a wide-spectrum contact and stomach insecticide with a quick 'knock-down' effect. Quinolphos 25% (*w/w*)-an organo-phosphorus acid ester. It is widely used in agricultural practices in India.

| | |
|---|---|
| CAS no. | 26124 |
| Chemical name | Diethquinalphion |
| Trade name | Ekalux Quinalphos 25 EC |
| Chemical formula | $C_{12}H_{15}N_2O_3PS$ |
| Chemical nature | Organothiophosphate |
| Molecular weight | 298.3 g/mol |

2.2.2. Impala (EC 55%): Chlorpyrifos (OP) EC 50%; Cypermethrin (CP/SP) EC-5%

Impala EC-55% (Syngenta India Limited, Pune, India) is a combination of Chloropyrifos EC-50% and Cypermethrin EC-5%.

| | | |
|---|---|---|
| CAS no. | 2921-88-2. | 52315-07-8 |
| | 0,0-diethyl 0-(3,5,6-trichloro-2-pyridinyl)-phosphorothioate | |
| Chemical name | (R,S)-alpha-cyano-3-phenoxybenzyl(1RS)-cis,trans-3-(2,2-dichlorovinyl)-2,2-dimethylcyclopropane-carboxylate | |
| Trade name | Dursban | Auzar 25 EC |
| Chemical formula | $C_9H_{11}Cl_3NO_3PS$ | $C_{22}H_{19}Cl_2NO_3$ |
| Chemical nature | Chlorinated organophosphate (OP) | Carboxylic ester |
| Molecular weight | 350.59 g/mol | 416.3 g/mol |

2.2.3. Neemstar–Azadirachtin

Neemstar is a neem-based natural pesticide. Azadirachtin (3%), a neem derivative (*Azadirachta indica*), acts as a repellent and has antifeedant properties. Neem seed includes a number of compounds (azadirachtin, sulfur, toluene, and fatty acids) that suppress the population of insect pests. Control of a variety of insects, particularly Bollworms, Aphids, Jassids, Thrips, Whiteflies, Leaf folders, Pod borer, Fruit borer, Leaf hopper, and Diamondback moth in Cotton, Rice, Pigeon pea, Chickpea, Safflower, Okra, Cauliflower, Cabbage, and Tomato.

| | |
|---|---|
| CAS no. | 11141-17-6 |
| Chemical name | Azadirachtin |
| Trade name | Neemstar |
| Chemical formula | $C_{35}H_{44}O_{16}$ |
| Chemical nature | Natural |
| Molecular weight | 720.721 g/mol |

*2.3. Animal System: Mystus keletius*

*Mystus keletius* (Catfish) was chosen for this study due to its increasing prevalence and usefulness as a subject for toxicology testing. This fish species also demonstrates an excellent ability to adapt to changing environmental conditions, as well as tolerate laboratory conditions [35]. Fish were caught from ponds at the Thirupalai Kamma, Madurai, which is located at 9°58′34.5 North latitude and 78°08′44.4 East longitude. All *Mystus keletius* fingerlings used in this study ranged between 3.6–10 g and 4–8 cm in length. Prior to the start of the research, fish were gathered and housed in big circular plastic pools

for 2 months. Prior to testing, animals were acclimated for 7 days in glass aquaria with a compressed air supply in freshwater (ambient temperature 24.8–29.5 °C). Fish were fed pelleted feed once daily for 24 h prior to the start of the studies.

### 2.4. Acute Toxicity

The $LC_{50}$ result was determined using a semi-static acute toxicity bioassay. The sample solution of technical grade chlorpyrifos was produced utilizing acetone as a 5% stock solution and diluted as needed. The fish were separated into six groups of six fish each. Animals were kept in 20 L water-containing plastic tubs without food and aeration for the duration of the 96 h acute toxicity trial. To avoid the impact of sunshine and pesticide volatilization from surface water, the tubs were closed completely. Group 1 animals were subjected to chlorpyrifos at doses ranging from 1000 to 25 µg/L. Chlorpyrifos was dissolved in acetone. Simultaneously, controls receiving acetone (Group 2) in the same volume as the pesticide-treated group were kept to determine the mortality rate, if any, due to the vehicle impact. Groups 1 and 2 were studied for 96 h in conjunction with a control group that did not receive either acetone or pesticide [36]. Probit analysis was used to determine the mortality [37], and the $LC_{50}$ (the chlorpyrifos concentration at which 50% death was recorded) was calculated.

### 2.5. Hematological Analysis

Fish (3 fish from each of 3 experimental and control aquaria) were cold anesthetized and sacrificed via spinal dislocation after 7, 14, 21, and 28 days of exposure. Blood samples were obtained from all these fish's caudal veins. Hematological values were determined using established procedures. Sahli's acid haematin method was used to determine hemoglobin (Hb), as described by Darmady and Davenport [38]. Neubauer's improved hemocytometer was used to count red blood cells (RBC) and white blood cells (WBC), utilizing Hyem's and Turk's solution as diluting fluids, as described by Darmady and Davenport [38]. The hematocrit/packed cell volume (PCV) was determined using Snieszko's microhematocrit technique [39]. The mean corpuscular volume (MCV), mean corpuscular hemoglobin (MCH), and mean corpuscular hemoglobin concentration (MCHC) derived hematological indicators were determined using standardized methods [40]. MCV was measured in femtoliters using the formula = PCV/RBCX10, MCH was computed in picograms using the formula = Hb/RBCX10, and MCHC was calculated using the formula = (Hb in 100 mg blood/PCV) × 100. Serum glucose was determined using the Cooper et al. [41] method, serum protein was determined using the Lowry et al. [42] method, and total serum cholesterol and HDL were determined using the Rifai et al. [43] method.

*Mystus keletius* (3.45 ± 0.74 g) were exposed to a sub-lethal concentration of pesticides (Ekalux (EC-25%); Impala (EC-55%); and NeemStar (EC-15%)) separately at room temperature. As a control, a group of fish was raised in a medium devoid of pesticides. During the exposure, fish were fed pellet food manufactured artificially once every day for two hours. The remaining food was removed after feeding. In all the experiments, the medium was changed every day. The experiments were conducted for a period of 28 days. Three fish from each control and experimental group were cold anesthetized and sacrificed on the seventh, fourteenth, twenty-first, and twenty-eighth days after exposure. The dissected tissues were cleaned with 0.89% ice-cold sodium chloride saline and transferred to suitably labeled sterilized polythene bags, which were subsequently stored in a laboratory freezer (−20 °C) for the subsequent investigation. The sample was homogenized in a mortar with 3 mL of ice-cold saline (0.89% NaCl) solution. To obtain a clear saline precipitate, the samples were centrifuged at 4000 rpm for 45 min at 5 °C in a chilled centrifuge. The quantification of enzymes, such as succinate dehydrogenase (SDH), malate dehydrogenase (MDH), and glutamate dehydrogenase, was performed using an aqueous muscle extract in ice-cold saline (GDH).

### 2.6. Succinate Dehydrogenase (SDH)

SDH activity was estimated by the method of Nachlas et al. [44]. Liver and muscle tissues of the control and treated fish were homogenized for 1 to 3 min with a mortar. A stock homogenate of 5 mg/L (wet weight) was prepared in 0.1M phosphate buffer, pH-7.7. The coarse particulate was separated using a clinical centrifuge for approximately 30 s, and the translucent precipitate was used as the standard enzyme extract. The solution was prepared by mixing 40 mM sodium succinate, 4 mM 2-n-iodophenyl)-3-(p-nitrophenyl)-5-phenyl tetrazolium chloride (INT), 100 mM potassium phosphate buffer (pH 7), and 0.5 mL supernatant extract. For 30 min, the reactive mixture was incubated at 37 °C. To bring the reaction to a halt, 5 mL glacial acetic acid was added. The resultant iodo-formation was extracted overnight in 5 mL of toluene at 5 °C. A spectrophotometer was used to determine its optical density at 495 nm using a toluene blank. The enzyme activity was expressed as µM of formazan/mg protein/h.

### 2.7. Malate Dehydrogenase (MDH)

The method of Aliko et al. [45] was adopted. To 5 mL of the pyrophosphate extract of 0.1 g of dried material, 1 mL of 0.2M solution of sodium malate, and 0.1% TTC were added and incubated at 45 °C for 30 min. The developed red color was read at 420 nm along with the artificially reduced TTC standards. Five replicates were used for each of the enzyme studies. The average of the five replicates is given in the respective tables. All the experiments were carried out at $28 \pm 1$ °C.

### 2.8. Glyceraldehyde Dehydrogenase (GDH)

A known weight (0.1 gm) of homogenate was extracted with 0.5 mL of 0.1 M sodium pyrophosphate buffer (pH-8.6) and centrifuged at 3000 rpm for 5 min. One mL of freshly made 0.1% TTC solution and 1 mL of 1 M glycerol were applied to the whole supernatant. For 30 min, the samples were incubated at 45 °C. The generated red color was recovered with 6 mL of toluene, and its intensity was determined using a spectrophotometer set to 440 nm in conjunction with an artificially decreased TTC reference. Values are expressed as mg of the TTC reduced/wet tissue/h [46].

The findings in this article are reported as mean and standard error. The two-tailed Student's t-test was used to determine statistical significance ($p < 0.05$) between the control and experimental groups.

### 3. Results

The effect of Ekalux on RBC content reduced from 1.43 million/mm$^3$ on day 7 to 1.18 million/mm$^3$ on day 28. Impala's value on RBC increased from 1.36 million/mm$^3$ on day 7 to 1.10 million/mm$^3$ on day 28, whilst Neemstar's value declined from 1.48 million/mm$^3$ on day 7 to 1.02 million/mm$^3$ on day 28 (Figure 1a).

A similar tendency was detected for hemoglobin (g/dL) and HC (%) concentrations, which was nearly the opposite of the trend reported for WBC concentrations. On day 28, the WBC count in fish blood increased from $17.65 \times 10^4$ cells to $19.33 \times 10^4$ cells in response to the pesticide. A similar trend was found for Impala, where the value increased from $18.20 \times 10^4$ cells on day 7 to $22.54 \times 10^4$ cells on day 28. Similarly, on day 7, a value of $17.8 \times 10^4$ cells was reported for Neemstar, which increased to $18.27 \times 10^4$ cells on day 28 in treated fish (Table S1 and Figure 1b).

On day 7, the Hb content of experimental fishes exposed to Ekalux was 6.59 g/dL; by day 28, it had fallen to 2.00 g/dL, indicating a declining trend. Similar patterns were observed for Impala and Neemstar, where the value in treated Impala fish was 4.03 g/dL on day 28 and decreased to 1.65 g/dL. In contrast, the response of Neemstar was 6.73 g/dL on day 7, with the lowest value of 2.75 g/dL on day 28 (Table S1 and Figure 1c).

Similar observations were recorded for each content in percentage in response to the pesticide Ekalux on day 7 at the value of 32.75%, reaching a value of 22.68% on day 28 with intermediate values on day 14 and 21, respectively. A similar trend was recorded for

Impala, where the initial value of 30.96% was recorded in the test fishes that had a final value of 19.85% on day 28 (Table S1 and Figure 1d), showing a declining trend in Hc content in the experimental fishes. All the values are depicted in the table, and their corresponding figures in the form of a graph for MCV, MCH, and MCHC are presented in Figure 1e–g. In general, values in the control group were constant during the period of the study. Statistical analysis of the summative data using two-way ANOVA is provided (Table S1) to indicate the significance of the *p*-value (<0.001) obtained in the study.

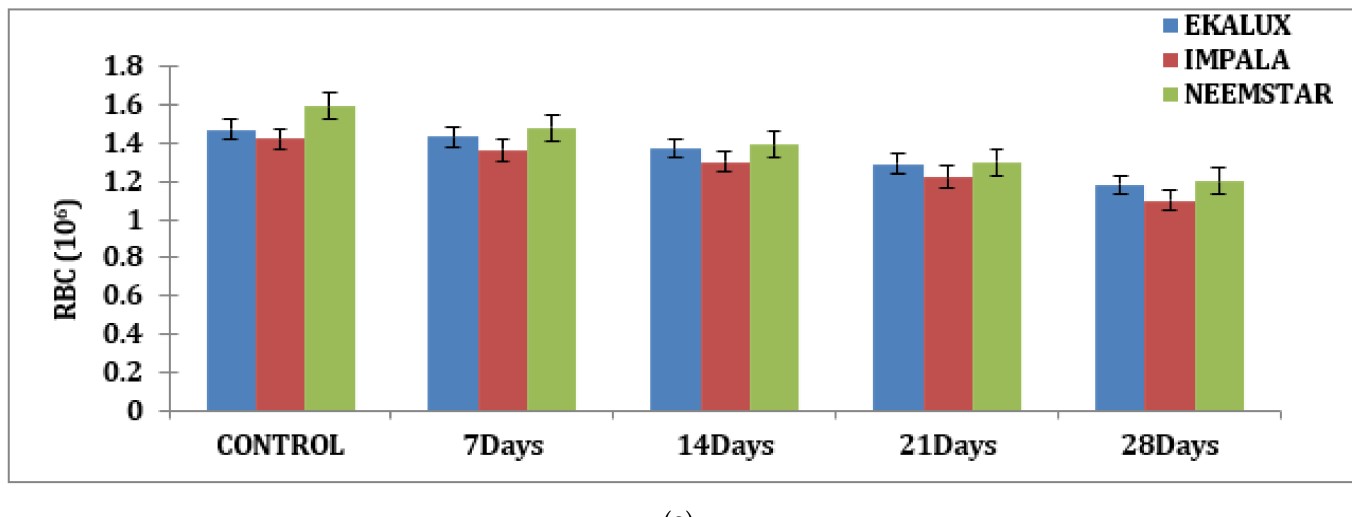

(**a**)

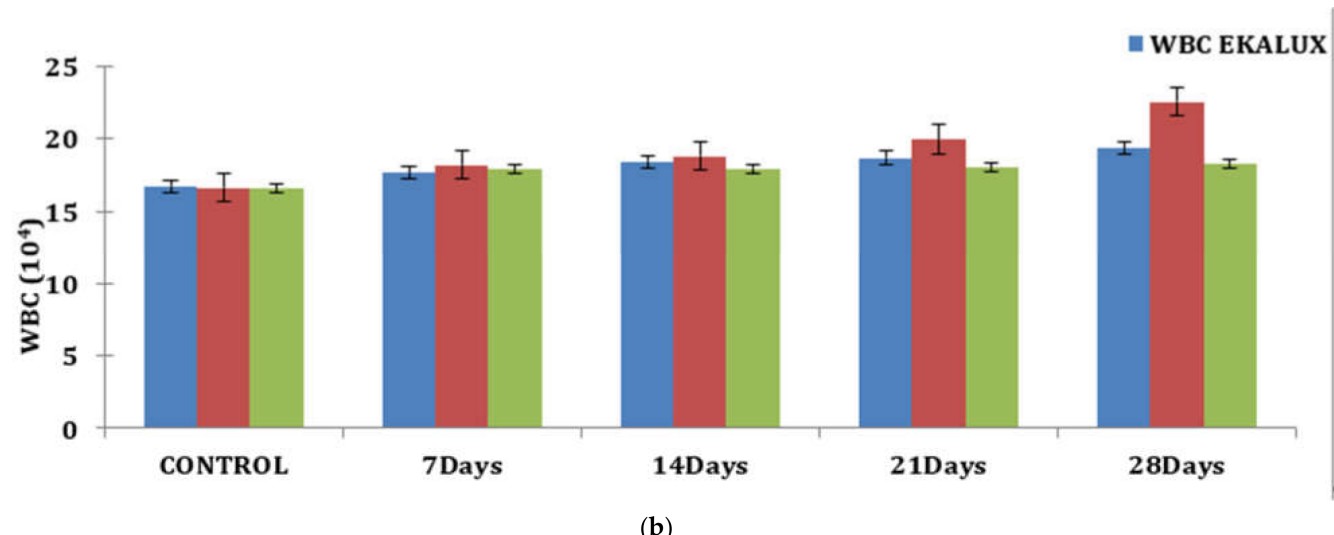

(**b**)

**Figure 1.** *Cont.*

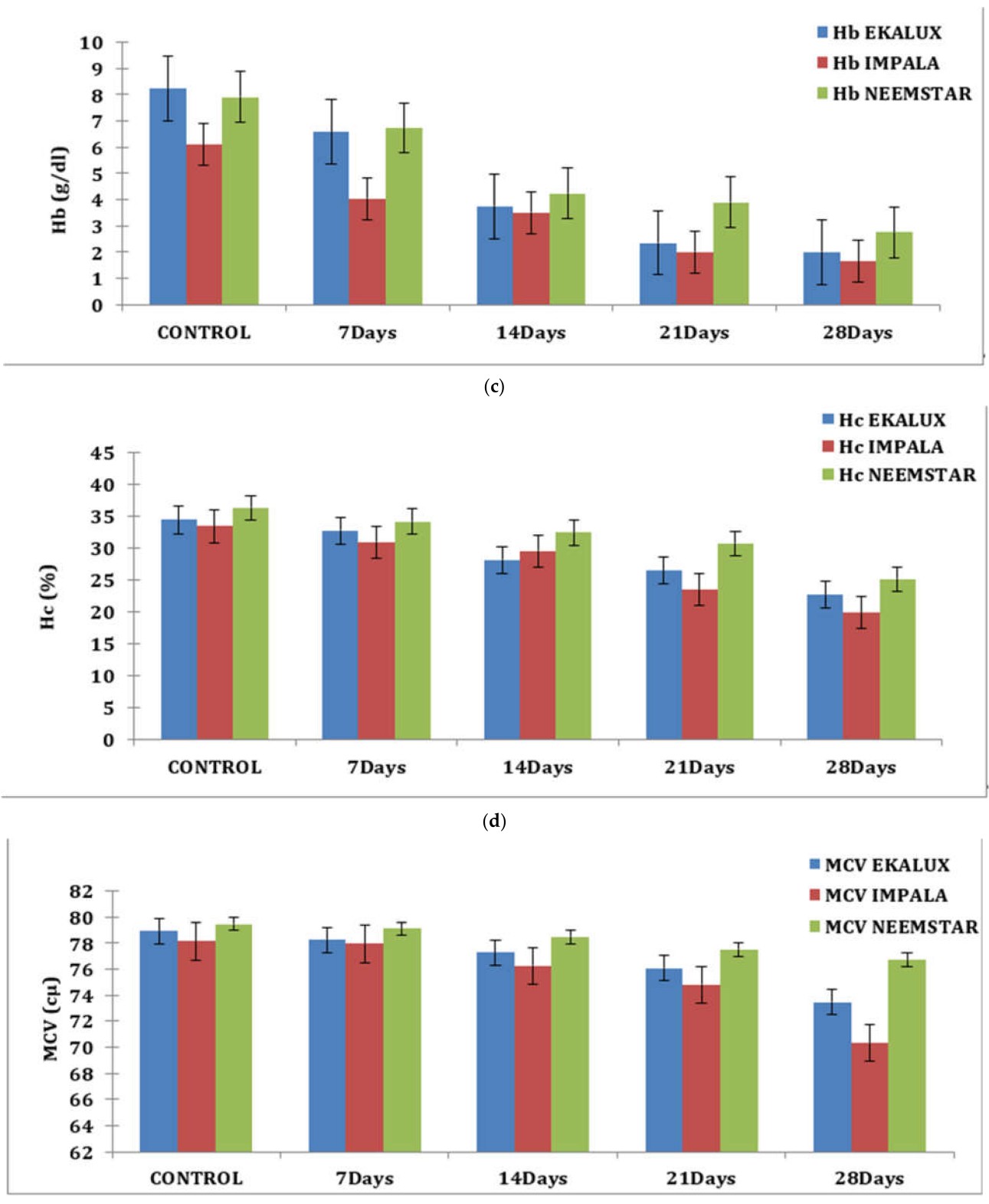

**Figure 1.** *Cont.*

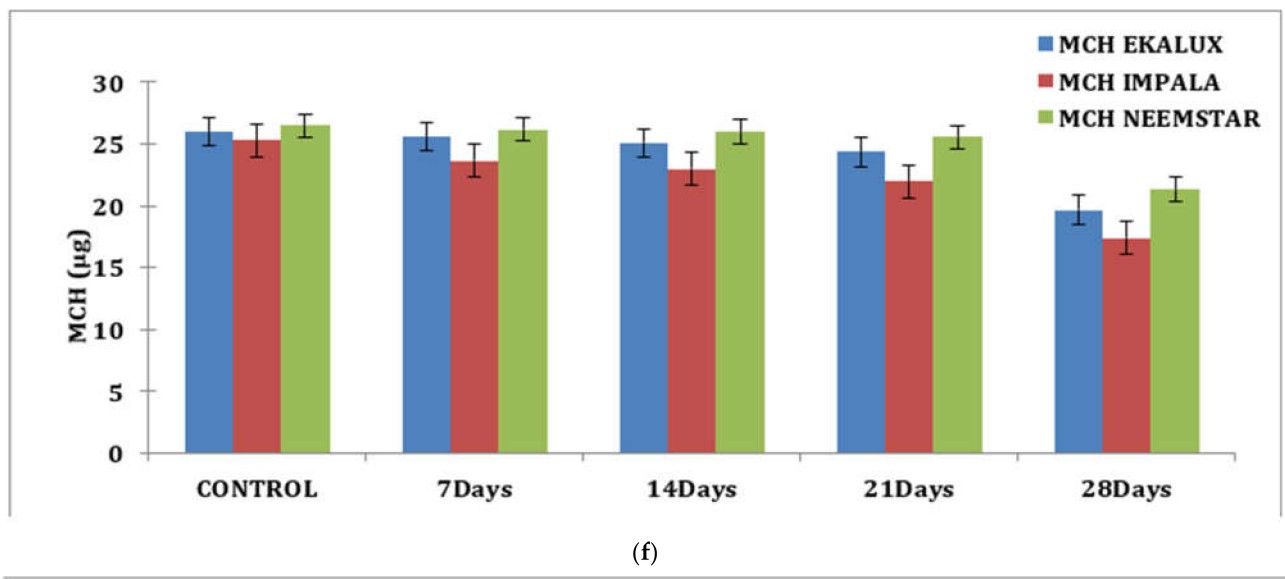

(f)

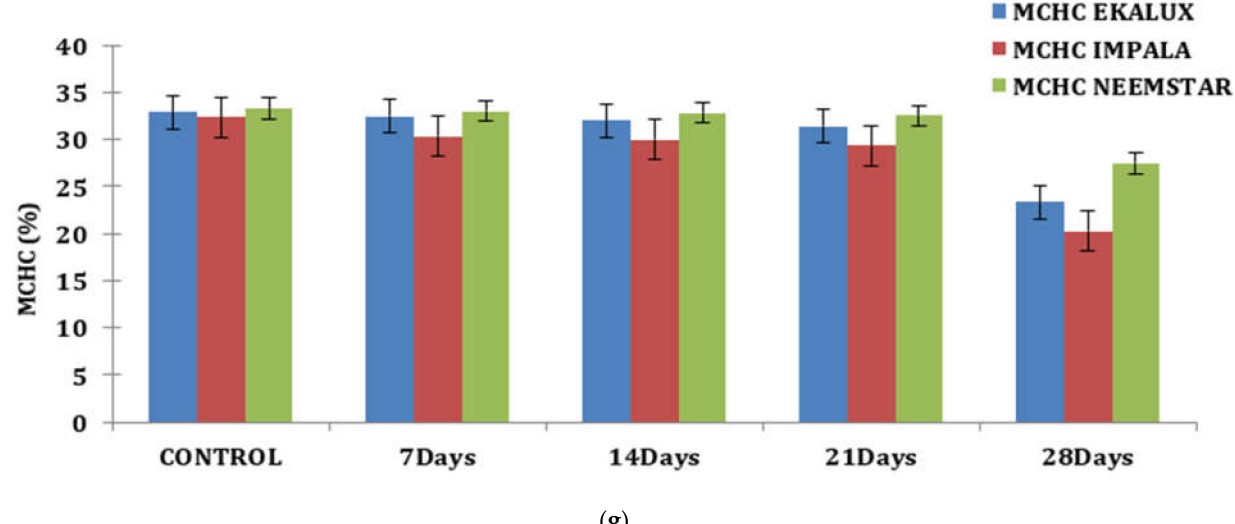

(g)

**Figure 1.** (**a**) Effect of selected pesticides on red blood cell levels in the blood of *Mystus keletius*. (**b**) Effect of selected pesticides on white blood cell levels in the blood of *Mystus keletius*. (**c**) Effect of selected pesticides on Hb (Hemoglobin) levels in the blood of *Mystus keletius*. (**d**) Effect of selected pesticides on Hc (Hemocrit) levels in the blood of *Mystus keletius*. (**e**) Effect of selected pesticides on MCV in the blood of *Mystus keletius*. (**f**) Effect of selected pesticides on MCH content in the blood of *Mystus keletius*. (**g**) Effect of selected pesticides on MCHC in the blood of *Mystus keletius*.

### 3.1. Effect of Ekalux on Respiratory Enzymes

The presence of the said pesticides in the aquatic medium significantly affected the tissue and the enzyme during the different time periods of exposure in the experimental animals with respect to its control. The effect due to the presence of a collapse in the brain is insignificant in the test animal during the exposure period from day 7 to 28. On day 7, the SDH activity in the test animals recorded a value of 0.50 mg reduced TTC/g/wet wt/h, which decreased 0.47 mg reduced TTC/g/wet wt/h on day 14 and reached a minimum value of 0.44 mg reduced TTC/g/wet wt/h on day 28 in the brain tissues (Table S2; Figure 2a). Likewise, a similar trend was observed for MDH activity. On day 7 in the brain tissue, 0.96 mg reduced TTC/g/wet wt/h was observed, followed by 0.86 mg reduced TTC/g/wet wt/h, 0.74 mg reduced TTC/g/wet wt/h, and 0.63 reduced TTC/g/wet wt/on day 14, 21, and 28 respectively (Table S2 and Figure 2a). On day 7, a significant increase compared with the control was observed for GDH activity, which increased to

0.58 mg reduced TTC/g/wet wt/h on day 14 and 1.07 mg reduced TTC/g/wet wt/h on day 21 and was maintained thereafter. The gills, liver, and muscle tissues also followed a similar pattern of changes when exposed to the pesticide Ekalux during the study period compared to the control (Table S2 and Figure 2d). All values acquired during the study were statistically significant at a *p*-value of less than 0.05, as determined by a statistical analysis of the data obtained (*p* < 0.05).

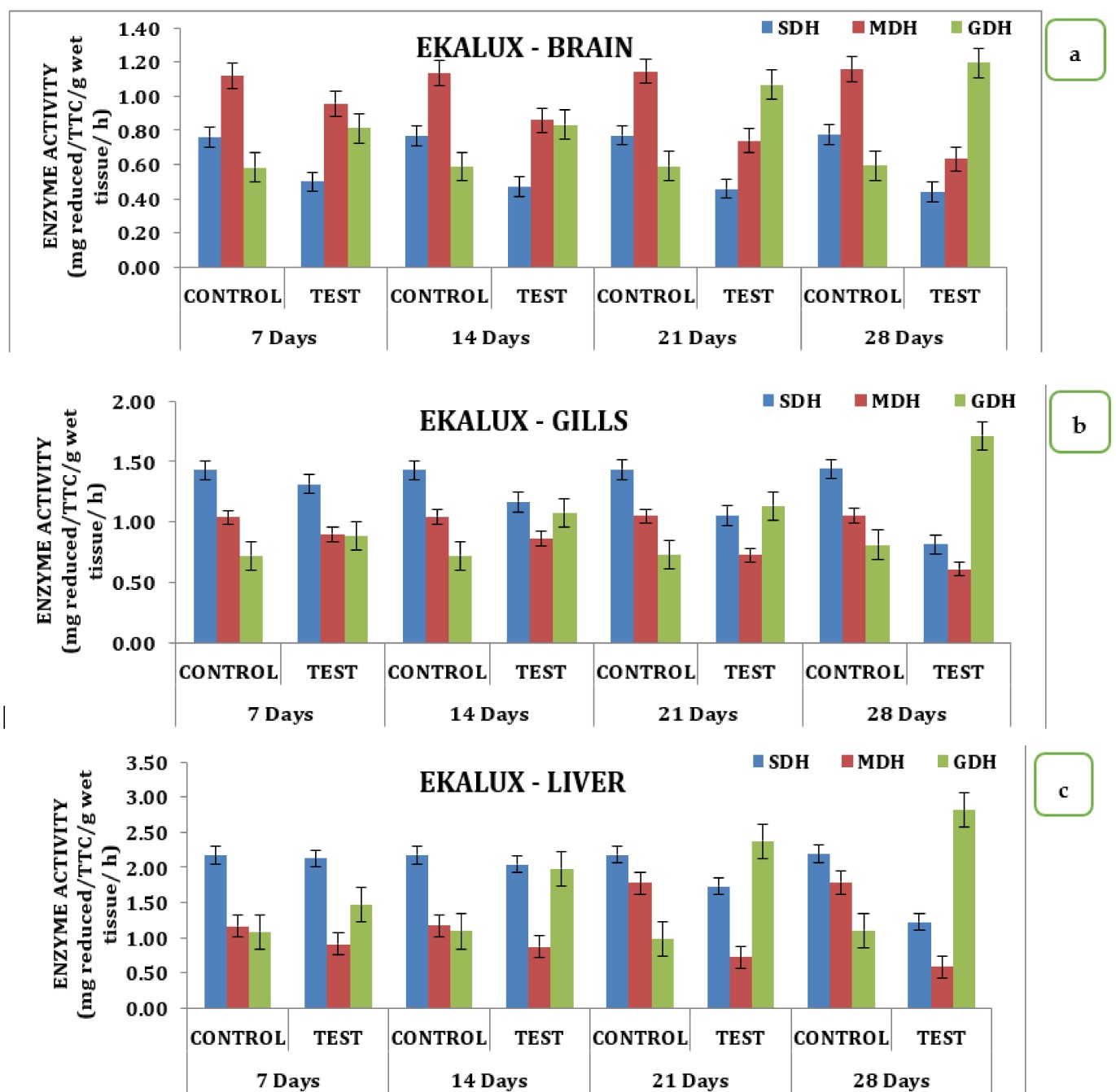

**Figure 2.** *Cont.*

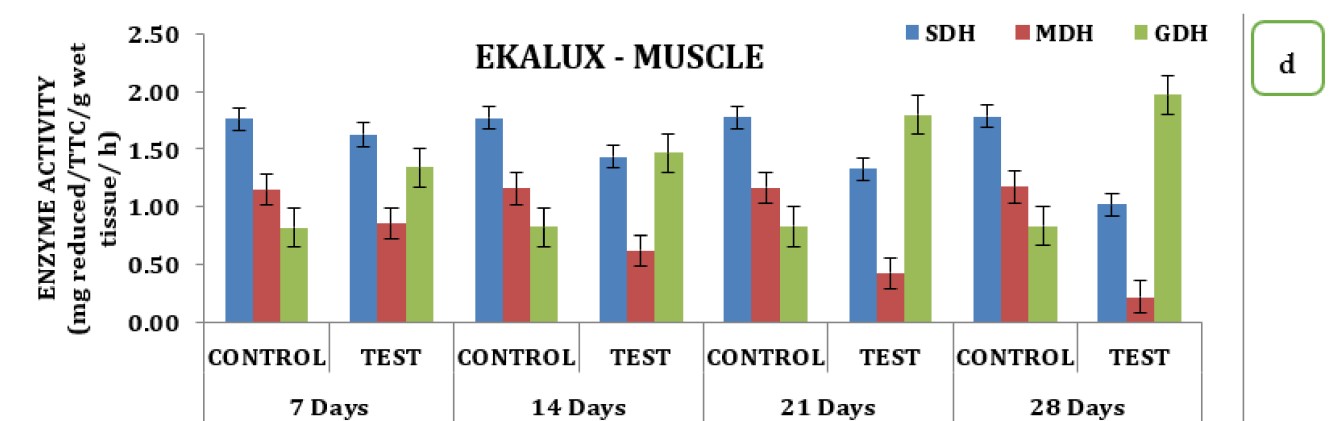

**Figure 2.** Effect of Ekalux on the activity of respiratory enzymes in different tissues of *M. keletius* (**a**) brain; (**b**) gills; (**c**) liver; (**d**) muscle.

### 3.2. Effect of Impala on Respiratory Enzymes

The effect of the pesticide Impala on brain, gill, liver, and muscle tissue over SDH, MDH, and GDH activity during different days of exposure had significant variations in the data. Analysis of the data indicates that in the brain, there was a sharp decline in SDH activity with its respective test medium on day 7 to a value of 0.42 reduced TTC/g/Wet wt to that declined on day 14 is 0.36 mg reduced TTC/g/wet wt/h and day 21 the value of 0.34 mg reduced TTC/g/wet wt/h that reach the minimum of 0.32 mg reduced TTC/g/wet wt/h on day 28 (Table S2 and Figure 3). Similarly, MDH activity also recorded a declining trend starting from 0.91 mg reduced TTC/g/wet wt/h on day 7 to a minimum of 0.65 mg reduced TTC/g/wet wt/h on day 28. Likewise, the activity exhibited a variation where there was a slight improvement observed on day 7 that with a continuous increase in activity and reached a maximum on day 28, indicating that the activity of GDH had an inverse relationship as a function of time in the brain tissue (Table S2 and Figure 3a). Analysis of different respiratory enzymes, namely SDH, MDH, and GDH in gill tissue in response to Impala exhibited wide variation in the activity during different exposure times on the seventh day. SDH activity in the gills recorded the value of 0.91 mg reduced TTC/g/wet wt/h in the experimental animal that for the reduced to 0.86 mg reduced TTC/g/wet wt/h on day 14, 0.72 mg reduced TTC/g/wet wt/h on day 21 and reached a minimum of 0.56 mg reduced TTC/g/wet wt/h on day 28. A similar trend was observed for MDH activity that had a value of 0.84 mg reduced TTC/g/wet wt/h on day 7 and reached a minimum value of 0.49 mg reduced TTC/g/wet wt/h on day 28 with intermediate values on 14th and 21st days respectively (Table S2 and Figure 3b). However, quite a different kind of trend was observed for GDH activity in the case of the brain. In the liver, where the activity rose from 1.09 mg reduced TTC/g/wet wt/h on day 7 to 1.77 mg reduced TTC/g/wet wt/h on day 28 with a gradual increase in the intermediate days in the liver stage activity, on day 7 had a value of 1.09 mg reduced TTC/g/wet wt/h that reached a maximum of 1.77 on day 28 with 1.12 mg reduced TTC/g/wet wt/h on day 14 and 1.55 mg reduced TTC/g/wet wt/h on day 21 of the treatment (Table S2 and Figure 3c). However, a quite opposite, declining trend was observed for the MDH activities. The MDH activity in the liver with the treatment to Impala had a differential response on the 7th and reached a minimum value of 0.77 mg reduced TTC/g/wet wt/h on the 28th day. Likewise, muscle tissue recorded a value of 1.41 mg reduced TTC/g/wet wt/h on day 7 that reached a maximum value of 0.68 mg reduced TTC/g/wet wt/h, depicting the declining trend in its response (Table S2 and Figure 3c). Likewise, MDH in muscle tissue had a value of 0.86 mg reduced TTC/g/wet wt/h on day 7 that reached a minimum value of 0.4 mg reduced TTC/g/wet wt/h on day 28, with intermediate values on day 21 and day 14, respectively. GDH activity exhibited an increasing trend with a value of 1.43 mg reduced TTC/g/wet wt/h on day 7, and a maximum of 2.15 mg reduced TTC/g/wet wt/h on day 28, with

intermediate values on day 14, and a value of 2.05 mg reduced TTC/g/wet wt/h on day 21 (Table S2 and Figure 3d). When analyzed statistically, the data indicate that all the values were significant at $p < 0.05$.

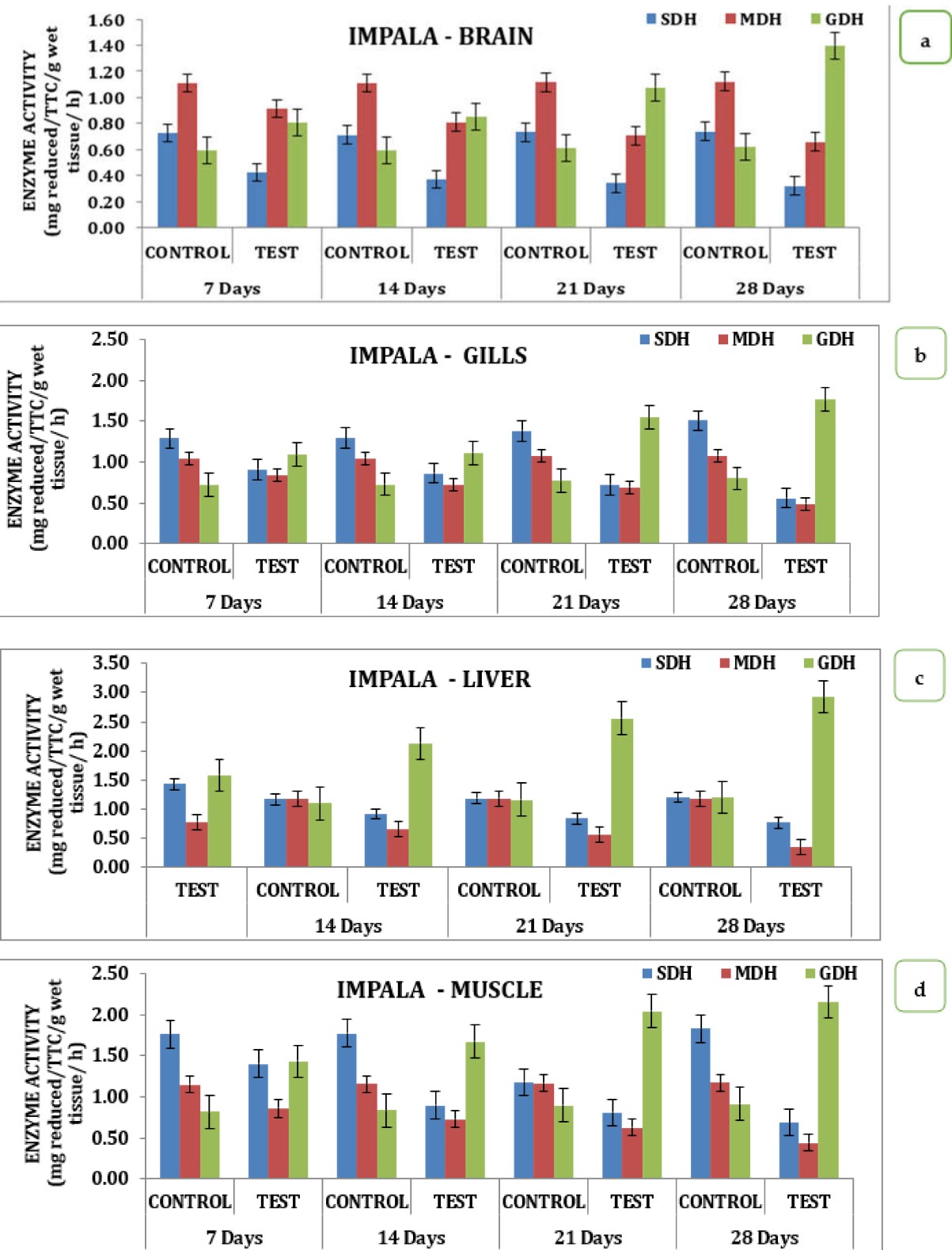

**Figure 3.** Effect of Impala on activity of respiratory enzymes in different tissues of *M. keletius* (**a**) brain; (**b**) gills; (**c**) liver; (**d**) muscle.

### 3.3. Effect of Neemstar on Respiratory Enzymes

The effect of Neemstar on different tissue systems over the treatment period in the experimental animals had interesting changes in the activity pattern of enzymes from day 0 to day 28 of treatment. On the 7th day of treatment, SDH activity was recorded to be 0.6 mg reduced TTC/g/wet wt/h, which reached 0.5 mg reduced TTC/g/wet wt/h on the 28th day, with 0.57 mg reduced TTC/g/wet wt/h on the 14th day, and 0.55 mg reduced TTC/g/wet wt/h on the 21st day. A similar trend was observed for MDH activity, which had a minimum value of 0.57 mg reduced TTC/g/wet wt/h on the 28th day with values corresponding to 0.74, 0.89, 0.98 mg reduced TTC/g/wet wt/h on 21st, 14th and 7th day, respectively. GDH activity showed an increasing trend starting from a value of 0.77 mg reduced TTC/g/wet wt/h on the 7th day and increased to 0.82 mg reduced TTC/g/wet wt/h on the 14th day; 1.04 mg reduced TTC/g/wet wt/h on 21st day and reached a maximum of 1.19 mg reduced TTC/g/wet wt/h on the 28th day. Data indicates a declining trend was observed for SDH and MDH activities, and an inverse relationship was exhibited by GDH activity in brain tissues (Table S2 and Figure 4a). When analyzed for its activity, SDH, MDH, and GDH showed a differential value in gill tissue. SDH activity on the 7th day had a value of 1.40 mg reduced TTC/g/wet wt/h on the 7th day that reached a value of 0.99 on the 28th day. A similar trend was observed for MDH activity with a value of 0.49 on day 7 that reached a minimum of 0.59 on the 28th day (Table S2 and Figure 4b). In the liver, MDH and SDH activity exhibited wide variation in activity. SDH on the 7th day in the experimental animals recorded a value of 2.1 that reached a value of 1.4 mg reduced TTC/g/wet wt/h on the 28th day with intermediate values on the 14th and 21st days, respectively. A similar trend was observed for MDH activity in the liver with a value of 0.99 TTC/g/wet wt/h on the 7th day, reaching a minimum of 0.34 TTC/g/wet wt/h on the 28th day (Table S2 and Figure 4c). A similar trend was observed for three different enzymes studied in the muscle tissue in response to Neemstar on different days of exposure. SDH recorded values of 1.69 TTC/g/wet wt/h that reached a minimum of 1.1 TTC/g/wet wt/h on the 28th day of treatment. Correspondingly, a declining trend was observed for mph activity that had a value of 0.86 on the 7th day in the test animals that reached a minimum of 0.28 on the 28th day. As far as the MDH activities were concerned, the GDH activity showed an increasing trend in the muscle tissue that had values of 1.34 TTC/g/wet wt/h on the 7th day and reached a maximum of 1.56 mg reduced TTC/g/wet wt/h on the 28th day (Table S2 and Figure 4d). Statistical analysis of the data indicates that all the values were statistically significant at $p < 0.05$ level.

TCP levels in liver tissue and bile, which increased in a dose–response fashion, also revealed a significant degree of metabolism. The presence of the aforementioned pesticides in the aquatic medium had a substantial effect on the tissue and enzyme in the experimental animals over a range of time periods of exposure in comparison to the control. The effect of the collapse on the brain is negligible in the test animals during the 7th to 28th day of exposure. On day 7, the SDH activity in the test animal was 0.50 mg reduced TTC/g/wet wt/h, declined to 0.47 mg reduced TTC/g/wet wt/h on the 14th day, and finally achieved a minimum of 0.44 mg reduced TTC/g/wet wt/h on the 28th day in the brain tissues.

The effect of Impala on brain, gill, liver, and muscle tissue over SDH, MDH, and GDH activity varied significantly across days of exposure. Analysis of the data indicates that in the brain, there was a sharp decline in SDH activity with its respective test medium on the 7th day from a value of 0.42 mg reduced TTC/g/wet wt/h to that for the 14th day to 0.36 mg reduced TTC/g/wet wt/h and the 21st day was 0.34 mg reduced TTC/g/wet wt/h that reached a minimum of 0.32 mg reduced TTC/g/wet wt/h on the 28th day. Similarly, MDH activity decreased from 0.91 on the 7th day to 0.65 on the 28th day. Additionally, the activity varied, with a slight improvement observed on the 7th day followed by a continuous increase in activity to reach a maximum on the 28th day, indicating that GDH activity had an inverse relationship as a function of time in the brain tissue.

Analysis of different respiratory enzymes, namely SDH, MDH, and GDH in gill tissue in response to Impala exhibited wide variation in the activity during different exposure

times. On the 7th day, SDH activity in the gills recorded a value of 0.91 in the experimental animals that reduced to 0.86 on the 14th day 0.72 mg reduced TTC/g/wet wt/h on the 21st day and reached a minimum of 0.56 on the 28th day. A similar trend was observed for MDH activity that had a value of 0.84 on the 7th day and reached a minimum value of 0.49 on the 28th day with intermediate values on the 14th and 21st days, respectively.

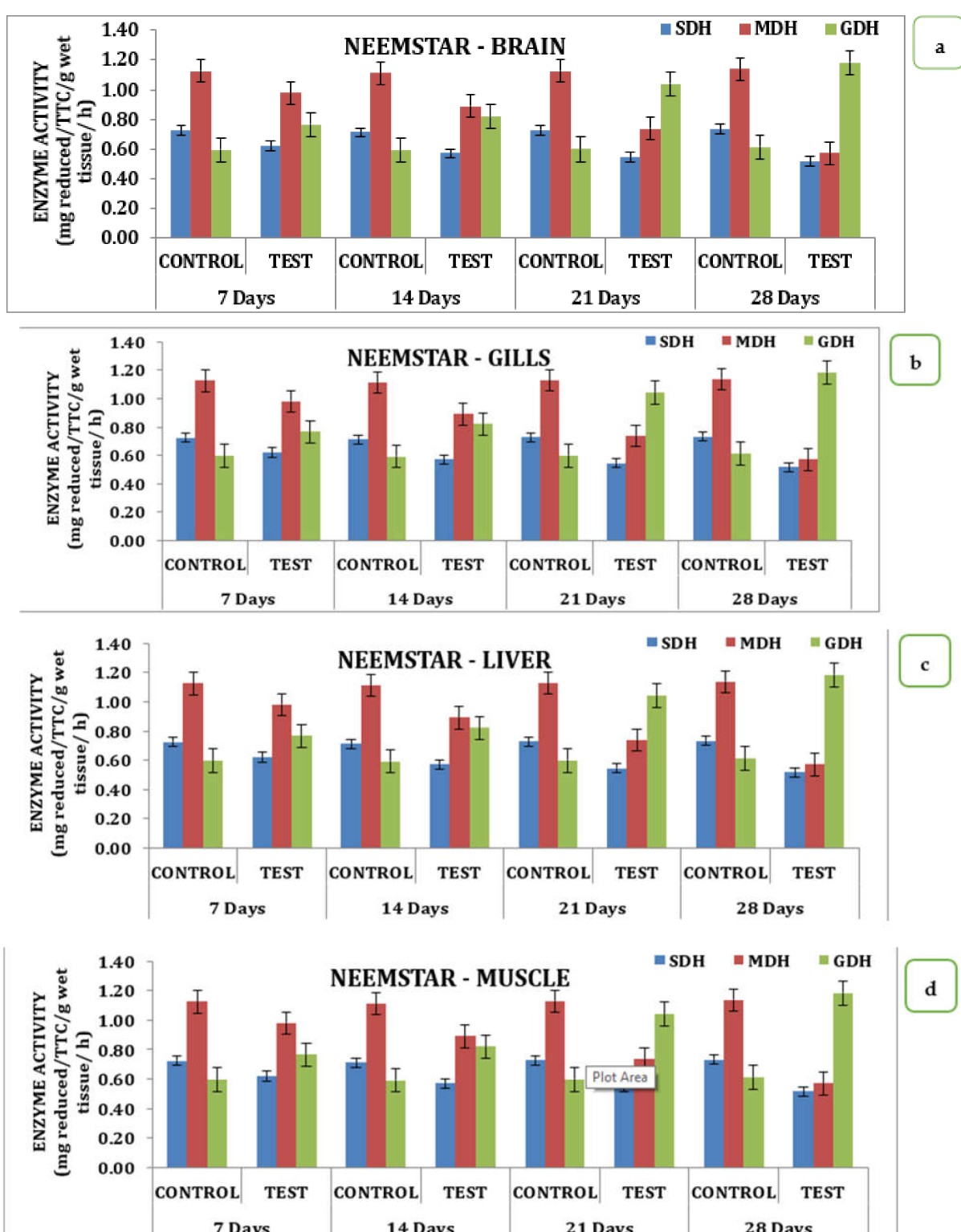

**Figure 4.** Effect of Neemstar on activity of respiratory enzymes in different tissues of *M. keletius* (**a**) brain; (**b**) gills; (**c**) liver; (**d**) muscle.

However, quite a different trend was observed for GDH activity in the case of the brain. In the liver, where the activity rose from 1.09 on the 7th day to 1.77 on the 28th day with a gradual increase in the intermediate days in the liver stage activity, on the 7th had a value of 1.09 that reached a maximum of 1.77 on the 28th day with 1.12 on 14th day and 1.55 on the 21st day of the treatment. However, quite the opposite declining trend was observed for the MDH activities. The MDH activity in the liver with the treatment to Impala had a differential response on the 7th and reached a minimum value of 0.77 on the 28th day. Likewise, muscle tissue recorded a value of 1.41 mg reduced TTC/g/wet wt/h on the 7th day that reached a maximum value of 0.68, depicting the declining trend in its response.

Likewise, MDH in muscle tissue had a value of 0.86 on the 7th day that reached a minimum value of 0.4 mg reduced TTC/g/wet wt/h on the 28th day with intermediate values on the 21st and 14th day, respectively. GDH activity exhibited an increasing trend with a value of 1.43 on the 7th day and a maximum of 2.15 on the 28th day, with intermediate values on the 14th day and a value of 2.05 mg reduced TTC/g/wet wt/h on the 21st day.

The effect of Neemstar on different tissue systems over the treatment period in the experimental animals had interesting changes in the activity pattern of enzymes from 0 to 28th days of treatment. On the 7th day, the SDH activity recorded a value of 0.6 that reached a value of 0.5 on the 28th day, with 0.57 on the 14th day and 0.55 on the 21st day. A similar trend was observed for MDH activity that had a minimum value of 0.57 on the 28th day with values corresponding to 0.74, 0.89, and 0.98 on the 21st, 14th, and 7th day, respectively. GDH activity showed an increasing trend starting from a value of 0.77 on the 7th day, increasing to 0.82 on the 14th day, 1.04 on the 21st day, and reaching a maximum of 1.19 on the 28th day. Data indicates that a declining trend was observed for SDH and MDH activities and an inverse relationship was exhibited by GDH activity in brain tissues.

When analyzed for its activity, SDH, MDH, and GDH showed a differential value in gill issue. SDH activity on the 7th day had a value of 1.40 mg reduced TTC/g/wet wt/h on that reached a value of 0.99 on the 28th day. A similar trend was observed for MDH activity with a value of 0.49 on the 7th day that reached a minimum of 0.59 on the 28th day. In the liver, MDH and SDH activity exhibited wide variation. SDH on the 7th day in experimental animals recorded a value of 2.1 that reached a value of 1.4 TTC/g/wet wt/h on the 28th day with intermediate values on the 14th and 21st day, respectively. A similar trend was observed for MDH activity in the liver, with a value of 0.99 on the 7th day. It reached a minimum of 0.34 on the 28th day.

A similar trend was observed for three different enzymes studied in the muscle tissue in response to Neemstar on different days of exposure. SDH recorded a value of 1.69 that reached a minimum of 1.1 on the 28th day of treatment. Correspondingly, a declining trend was observed for mph activity that had a value of 0.86 on the 7th day in the test animals that reached a minimum of 0.28 on the 28th day. As far as the MDH activities are concerned, the GDH activity showed an increasing trend in the muscle tissue that had a value of 1.34 on the 7th day and reached a maximum of 1.56 mg reduced TTC/g/wet wt/h on the 28th day.

Analysis of respiratory enzymes, namely SDH, MDH, and GDH in brain, gill, liver, and muscle tissue in response to Ekalux, Impala, and Neemstar exhibited wide variation in the activity during different exposure times on the 7th day. The overall pattern of pesticide activity of enzyme recorded a maximum for Impala, followed by Ekalux and Neemstar, indicating that Impala had a toxic effect on selected tissues when compared to the other two pesticides.

## 4. Discussion

Stress-induced hematological alterations in fish are type, magnitude, and time-dependent [47]. Hematological examination is a technique that is frequently used to determine the physiological wellbeing and health of fish [48,49]. The outcomes of hematological assessments in fish are influenced by a variety of intrinsic and extrinsic factors, as

well as the fish's stress level [50], the method of blood sampling [51], preanalytical factors such as anticoagulant use [52], the temperature and time of blood storage, and analytical procedures such as the type of diluent used [53], and the recorded information provided by the analysis of blood cells. Measured hematological parameters can fluctuate due to environmental conditions, notably food, water quality, anxiety, and pathogenic organisms, most notably the impact of xenobiotics on aquatic animal systems. All three different selected pesticides, namely Ekalux, Impala, and Neemstar were evaluated for their effects on the blood parameters, that is, RBC, WBC hemoglobin content MCV, MCH, and MCHC.

The erythrocyte count (RBC), a critical diagnostics indicator, is affected by a variety of ecological parameters, for example, water temperature [54]. Additionally, it can be modified by a variety of biological parameters, including fish behavior, aging, gender, nutritional requirements, and reproductive potential, and can vary between populations of the same species. While brief periods of stress might result in an increase in WBC, persistent and/or severe stress frequently results in leukopenia [55]. Stress may have an effect on leukocyte count by causing a change in the proportion of lymphocytes to neutrophils and monocytes [56]. According to Zaragoza et al. [57], both acute and chronic thermal stress have a substantial effect on red and white blood cell parameters in a manner that varies according to the aquatic body's temperature. Additionally, stress-related changes in blood parameter values are reflected in changes in biochemical parameters, including an increase in cortisol, glucose, and lactate levels [58].

Suppression of enzymatic activity in fish is a well-documented impact of OPs such as CPF [59–61]. Biotransformation following phase I oxidation and release of CPF-oxon into the blood was previously demonstrated, as all three exposure doses decreased enzyme activity considerably, and the parent molecule was released into the blood [62,63].

Glusczak et al. [36] examined glyphosate's acute impact on physiological and biochemical markers in silver catfish (Rhamdia quelen). Similarly, on day 7, 0.96 mg reduced TTC/g/Wet wt/h was detected in brain tissue, followed by 0.86, 0.74, and 0.63 mg reduced TTC/g/Wet wt/h on days 14, 21, and 28. On day 7, a similar significant increase in GDH activity was observed relative to the control, which increased to 0.58 mg reduced TTC/g/Wet wt/h on day 14 and 1.07 mg reduced TTC/g/wet wt/h on day 21, where it remained. The gills, liver, and muscle tissues also showed a similar pattern of changes when exposed to the pesticide Ekalux during the study period relative to the control.

Numerous studies have demonstrated histopathological destruction in mosquito fish (*G. affinis*) exposed to deltamethrin [64], yellow perch and goldfish (*C. auratus*) exposed to oil sands [65], yellow perch (*P. flavescens*) exposed to naphthenic acid [65], carp (*C. carpio*) exposed to deltamethrin [64], and rainbow trout (*O. mykiss*) exposed to maneb and carbaryl [66,67]. Significant increases due to pesticide exposures in the levels of RBC (Abamectin, Chlorpyrifos, Cypermethrin, Deltamethrin, Dichlorvos, Dimethoate, Fipronil, Lambda-cyhalothrin, Paraquat), HGB (Abamectin, Cypermethrin, Fipronil, λ-cyhalothrin, Paraquat), and HCT (Abamectin, Cypermethrin, Deltamethrin, λ-cyhalothrin, Paraquat) differ from findings of some earlier investigations using comparable pesticides, indicating the specific level of contradictions in hematological responses [68].

Chinnadurai et al. [69] investigated the hemodynamic alterations and oxygen consumption of *Oreochromis mosambicus* subjected to sublethal lead concentrations. For example, some investigators found a considerable drop in RBC and HGB levels following exposure to pesticides and other toxicants [70–73]. These alterations reported in this study can be related to direct responses to structural damage to the RBC membrane, which results in hemolysis, as well as the following requirement to rapidly create replacement blood cells to avoid anemia. The present study assessed the effects of three different pesticides, namely Ekalux, Impala, and Neemstar, on blood parameters such as RBC, WBC hemoglobin content, MCV, MCH, and MCHC.

The effect of pesticides on RBC content was 1.43 million/mm$^3$ on the 7th day and decreased to 1.18 million/mm$^3$ on the 14th and 21st days, respectively. A similar trend was observed for Impala on RBC content, which started at 1.36 million/mm$^3$ on the 7th day

and decreased to 1.10 million/mm$^3$ on the 28th day, while Neemstar value decreased from 1.59 million/mm$^3$ on the 7th day in control to 1.02 million/mm$^3$ on the 28th day. Comparable results were obtained by Magare and Patil [74] where the effect of pesticides on oxygen consumption, in addition to red blood cell count and metabolites of *Puntius ticto*.

According to Amaeze et al. [75], exposure of *C. gariepinus* to sub-lethal concentrations of 10 selected pesticides resulted in a substantial rise in the fish's WBC count, exception of chlorpyrifos, which resulted in a substantial reduction in the fish's WBC count, which contradicted the present study's results. A significant rise in mortality was recorded in fish exposed to lambda cyhalothrin, indicating the pesticide's high toxicity to *C. gariepinus*. Increased WBC count is associated with increased antibody production, which aids in the survivability of fish subjected to sublethal pesticide concentrations [76]. This reaction has also been reported in the very same fish species following acute treatment with chlorpyrifos and DDforce [77], as well as in *Cyprinus carpio* following acute exposure to phenithrotion and dichlorvos. This could be due to the re-release of white blood cells (WBC) from the spleen into the bloodstream in response to the toxic substance. As Muralidharan [78] suggests, some chemical substances, including insecticides, induce the production of antibodies as a result of their immunosuppressive effect. Hb content in the experimental fishes exposed to Ekalux on the 7th day recorded the value of 6.59 g/dL that reached the final value of 2.00 g/dL on the 28th day, indicating the declining trend. A similar trend was observed for Impala and Neemstar, where the value in the treated fishes was 4.03 g/dL that reached a value of 1.65 g/dL on the 28th day in Impala and 6.73 g/dL on the 7th day to a least value of 2.75 g/dL on the 28th day in response to the pesticide Neemstar.

Similar observations were recorded for each content in percentage in response to the pesticide Ekalux on the 7th day at the value of 32.75%, that reached a value of 22.68% on the 28th day with intermediate values on the 14th and 21st day, respectively. A similar trend was recorded for Impala, where the initial value of 30.96% was recorded in the test fishes that had a final value of 19.85% on the 28th day, showing a declining trend in Hc content in the experimental fishes. In general, values in the control were constant during the period of the study. Recently, Hong et al. [79] studied the changes in hematological and biochemical parameters and revealed the genotoxicity and immunotoxicity of neonicotinoids on Chinese rare minnows (*Gobiocypris rarus*).

## 5. Conclusions

In the present study, all three selected pesticides, namely Ekalux, Impala, and Neemstar, were evaluated for their effects on blood parameters RBC, WBC, Hb, Hc, MCV, MCH, and MCHC. Data indicate a decline in the resultant value for each of the parameters RBC, WBC, Hb, Hc, MCV, MCH, and MCHC evaluated with respect to the control toward all pesticides (Ekalux, Impala, and Neemstar) tested over a period of 28 days. The effect of pesticides on RBC content on the 7th day was 1.43 million/mm$^3$, that decreased to 1.18 million/mm$^3$ on the 14th and 21st days, respectively. A similar trend was observed for Impala and Neemstar on RBC. The overall pattern of pesticide activity of enzyme recorded a maximum for Impala, followed by Ekalux and Neemstar, indicating that Impala had a toxic effect on selected tissues when compared to the other two pesticides tested.

**Supplementary Materials:** The following supporting information can be downloaded at: https://www.mdpi.com/article/10.3390/su14159529/s1, Table S1: Effect of selected pesticides (ekalux, impala and neemstar) on changes in blood parameter in *M. keletius* exposed for 0–28 days; Table S2: Effect of Ekalux, Impala and Neemstar on activity of respiratory enzymes (mg reduced TTC/g/wet wt/h) in tissues of *M. keletius*.

**Author Contributions:** Conceptualization, A.B., S.R., R.J., P.B.; Methodology, A.B., S.R., K.N.; Validation, A.B., S.R., K.N., R.J.; Formal analysis, A.B., R.J., P.B., C.F.; Investigation, A.B., S.R., K.N.; Original draft preparation, A.B., S.R., C.P., R.J.; Writing–review and editing, C.P., R.J., P.B., C.F.; Supervision, R.J., P.B., C.F. All authors have read and agreed to the published version of the manuscript.

**Funding:** This research received no external funding.

**Institutional Review Board Statement:** Not applicable.

**Informed Consent Statement:** Not applicable.

**Data Availability Statement:** Not applicable.

**Acknowledgments:** The authors sincerely acknowledge the PG and Research Department of Zoology, Yadava College (Men), Madurai, Tamilnadu, for the support in performing this study. The authors gratefully acknowledge the support and cooperation of the Department of Botany, Government Arts College, Melur, and the Management of MGR College, Hosur, Tamil Nadu.

**Conflicts of Interest:** The authors declare no conflict of interest.

## Abbreviations

| | |
|---|---|
| $LC_{50}$ | Median lethal level, which killed 50% of the animals |
| ppm | Parts per million |
| °C | Degree Celsius |
| g | Gram |
| hr(s) | Hour(s) |
| mg | Milligram |
| min | Minutes |
| TTC | Triphenyl tetrazolium chloride |
| SDH | Succinic dehydrogenase |
| GDH | Glyceraldehyde dehydrogenase |
| TCA | Tricarboxylic Acid |
| *w/v* | Weight per volume |
| $\mu g \cdot L^{-1}$ | Microgram per liter |
| μ mol | Micromoles |
| °C | Degree Celsius |
| ANOVA | Analysis of variance |
| MCH | Mean Corpuscular Hemoglobin |
| MCHC | Mean Corpuscular Hemoglobin Concentration |
| MCV | Mean Corpuscular Volume |
| mg/L | Milligram per liter |
| N | Normality |
| NaCl | Sodium Chloride |
| MDH | Malate dehydrogenase |
| $p < 0.05$ | Probability is less than 0.05 level of significance |
| ppm | Parts per million |
| RBC | Red blood corpuscles |
| SD | Standard Deviation |
| SDH | Succinate dehydrogenase |
| WBC | White blood corpuscles |

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
