# Peer review of "Ecotoxicological Effects of Pesticides on Hematological Parameters and Oxidative Enzymes in Freshwater Catfish, Mystus keletius"

_sustainability, doi:10.3390/su14159529_

Round 1
Reviewer 1 Report
Some comments for authors to improve their manuscript and questions to answer and include as discussion in their manuscript:
- Line 48: Maybe it’s better to say “lethal impact” instead of “deadly impact”
- In introduction, authors, do not mention the neonicotinoid, pyrethroid and carbamates pesticides that also are used in large amounts in agriculture and may affect the aim of this study. Cypermethrin (see line 110) is a pyrethroid.
- Part 2 (Materials and Methods): the list of pesticides used is not in a proper manner presented. Please revise. Also provide all reagents used for this study.
- For lines 138-139 please provide references to support this statement.
- Why authors performed acute toxicity tests for chlorpyrifos? This information is already known by literature. And why not for the other pesticides?
- The way that authors present the design of this study must be clearer. It is not properly presented in part 2.
- Tables 1 and 2 have too many information and cannot be easily read. Maybe can be provided as supplementary material and authors present the effect of selected pesticides in a different way (i.e. figures, charts).
- Do Authors have rights to report the effects of specific pesticide brands?
- Why did not authors perform these tests on technical or analytical grade of these pesticide compounds?
- Why authors selected these specific pesticides?
- Bioethics approval?
Author Response
Reviewer #1: Some comments for authors to improve their manuscript and questions to answer and include as discussion in their manuscript,
|
Comment 1 |
: |
Line 48: Maybe it’s better to say “lethal impact” instead of “deadly impact” |
|
Response |
: |
Changed as per the suggestions of the reviewer |
|
|
|
|
|
Comment 2 |
: |
In introduction, authors, do not mention the neonicotinoid, pyrethroid and carbamates pesticides that also are used in large amounts in agriculture and may affect the aim of this study. Cypermethrin (see line 110) is a pyrethroid. |
|
Response |
: |
It was discussed in detail in the discussion section (Hong et al. 2018) |
|
|
|
|
|
Comment 3 |
: |
Part 2 (Materials and Methods): the list of pesticides used is not in a proper manner presented. Please revise. Also provide all reagents used for this study. |
|
Response |
: |
A separate section was included for pesticides and materials utilized. This section included a list of insecticides that were employed in this study |
|
|
|
|
|
Comment 4 |
: |
For lines 138-139 please provide references to support this statement. |
|
Response |
: |
Reference No. 35 (Verma AK, Prakash S. Impact of arsenic on haematology, condition factor, hepatosomatic and gastrosomatic index of a fresh water cat fish, Mystus vittatus. International Journal on Biological Sciences. 2019;10(2):49-54) is added to support the statement |
|
|
|
|
|
Comment 5 |
: |
Why authors performed acute toxicity tests for chlorpyrifos? This information is already known by literature. And why not for the other pesticides? |
|
Response |
: |
We conducted acute toxicity testing on chlorpyrifos because it is the most toxic component of Impala |
|
|
|
|
|
Comment 6 |
: |
The way that authors present the design of this study must be clearer. It is not properly presented in part 2. |
|
Response |
: |
The entire section has been improved in response to the reviewer's comments |
|
|
|
|
|
Comment 7 |
: |
Tables 1 and 2 have too many information and cannot be easily read. Maybe can be provided as supplementary material and authors present the effect of selected pesticides in a different way (i.e. figures, charts). |
|
Response |
: |
Tables 1 and 2 have been relegated to the supplementary materials area as Tables 1S and 2S, respectively |
|
|
|
|
|
Comment 8 |
: |
Do Authors have rights to report the effects of specific pesticide brands? |
|
Response |
: |
Due to the fact that the aforementioned brand of pesticides is widely used by farmers in the field and is readily available on the market, the issue of rights is moot |
|
|
|
|
|
Comment 9 |
: |
Why did not authors perform these tests on technical or analytical grade of these pesticide compounds? |
|
Response |
: |
We did not conduct these tests on technical or analytical grade pesticide chemicals because these are the pesticides utilised in local rice fields, and so researching their effect is more pertinent |
|
|
|
|
|
Comment 10 |
: |
Why authors selected these specific pesticides? |
|
Response |
: |
We chose these specific pesticides because they are commonly used in local rice fields, making their effects more relevant |
|
|
|
|
|
Comment 11 |
: |
Bioethics approval? |
|
Response |
: |
Not required as per Committee for the Purpose of Control and Supervision of Experiments on Animals (CPCSEA) regulations for this species |
Reviewer 2 Report
Ecotoxicological effects of pesticides on haematological parameters and oxidative enzymes in freshwater catfish, Mystus keletius
The authors investigated the effects of three pesticides on Mystus keletius. The article appears to be scientifically sound and interesting. For the most part, the organization and structure are good, but the manuscript is a little difficult to read as written. The article would be vastly improved by having it reviewed by a native English speaker or a language service.
Examples of grammar issues:
Line 20: … for a period of 7th, 14th, 21st and 28th days.
Consider: … for a period of 7, 14, 21 and 28 days.
Line 30 and line 31:… on 7th day. …on 28th day.
Consider: … on day 7
Consider: … on day 28
These needs adjusted throughout the entire manuscript.
Line 54: … the growing demand agro-industrial products.
Consider: … the growing demand for agro-industrial products.
Line 254-256: Sentences like this are awkward to read. There are no units on the values.
Consider something like: When the test organism was exposed to Impala a similar increasing trend was observed. The xxx value was 18.20 (units) on day 7, while on day 28 the value rose to 22.54 (units).
Lines 448-451: This sentence is awkward as written. I think there are multiple ideas in this sentence and needs to be split apart.
Line 525: … the recorded information of blood cells.
Not sure what this means.
Line 526-527: Hematological parameters are not susceptible, they fluctuate.
Consider something like: Measured hematological parameters can fluctuate due to environmental conditions…
Line 549-550: This is an example where values without units, and 14th, 21th, ect make the sentence awkward to read.
These should be adjusted throughout the manuscript.
Other considerations:
Key word: Please use key words that are not in title. All words in the title are already key words so its best to avoid them in the key words section.
Abstract: Typically, there are not abbreviations (example: packed cell volume (PCV)) in the abstract. Please remove.
Please make sure all scientific names are italicized: See lines 138, 142, 582, 586
Please make sure all unit spacing is consistent: See differences in line 143, 153, 164, 195, 202, 220, 226, ect.
Please spell out abbreviations if they are the first word in a sentence: See lines 185 and review manuscript.
Units: If there are units to the values, please add them after each value in the main body of the text.
Author Response
The authors investigated the effects of three pesticides on Mystus keletius. The article appears to be scientifically sound and interesting. For the most part, the organization and structure are good, but the manuscript is a little difficult to read as written. The article would be vastly improved by having it reviewed by a native English speaker or a language service.
Examples of grammar issues:
|
Comment 1 |
: |
Line 20: … for a period of 7th, 14th, 21st and 28th days. Consider: … for a period of 7, 14, 21 and 28 days |
|
Response |
: |
Changed as per the suggestions of the reviewer |
|
|
|
|
|
Comment 2 |
: |
Line 30 and line 31:… on 7th day. …on 28th day. Consider: … on day 7 Consider: … on day 28 These needs adjusted throughout the entire manuscript |
|
Response |
: |
Changes suggested by the reviewer has been carried out throughout the manuscript |
|
|
|
|
|
Comment 3 |
: |
Line 54: … the growing demand agro-industrial products. Consider: … the growing demand for agro-industrial products. |
|
Response |
: |
Modified as per the suggestions of the reviewer |
|
|
|
|
|
Comment 4 |
: |
Line 254-256: Sentences like this are awkward to read. There are no units on the values. Consider something like: When the test organism was exposed to Impala a similar increasing trend was observed. The xxx value was 18.20 (units) on day 7, while on day 28 the value rose to 22.54 (units). |
|
Response |
: |
The whole section was rewritten, and units were included throughout the manuscript |
|
|
|
|
|
Comment 5 |
: |
Lines 448-451: This sentence is awkward as written. I think there are multiple ideas in this sentence and needs to be split apart. |
|
Response |
: |
The whole section was rewritten as per the suggestions of the reviewer |
|
|
|
|
|
Comment 6 |
: |
Line 525: … the recorded information of blood cells. Not sure what this means. |
|
Response |
: |
It has been corrected as "recorded information provided by blood cell analysis." |
|
|
|
|
|
Comment 7 |
: |
Line 526-527: Hematological parameters are not susceptible, they fluctuate. Consider something like: Measured hematological parameters can fluctuate due to environmental conditions… |
|
Response |
: |
The sentence was rewritten as per the suggestions of the reviewer |
|
|
|
|
|
Comment 8 |
: |
Line 549-550: This is an example where values without units, and 14th, 21th, ect make the sentence awkward to read. These should be adjusted throughout the manuscript. |
|
Response |
: |
Changes suggested by the reviewer has been carried out throughout the manuscript |
|
|
|
|
|
Comment 9 |
: |
Other considerations: Key word: Please use key words that are not in title. All words in the title are already key words so its best to avoid them in the key words section |
|
Response |
: |
Keywords are now modified as suggested by the reviewer |
|
|
|
|
|
Comment 10 |
: |
Abstract: Typically, there are not abbreviations (example: packed cell volume (PCV)) in the abstract. Please remove. |
|
Response |
: |
Changes suggested by the reviewer has been carried out |
|
|
|
|
|
Comment 11 |
: |
Please make sure all scientific names are italicized: See lines 138, 142, 582, 586 |
|
Response |
: |
All the scientific names in the manuscript are now corrected |
|
|
|
|
|
Comment 12 |
: |
Please make sure all unit spacing is consistent: See differences in line 143, 153, 164, 195, 202, 220, 226, ect. |
|
Response |
: |
Consistent unit spacing is now verified throughout the manuscript |
|
|
|
|
|
Comment 13 |
: |
Please spell out abbreviations if they are the first word in a sentence: See lines 185 and review manuscript. |
|
Response |
: |
Changes have been made in response to the reviewer's comments, and a separate section on abbreviations has been included at the end of the manuscript |
|
|
|
|
|
Comment 14 |
: |
Units: If there are units to the values, please add them after each value in the main body of the text. |
|
Response |
: |
The units for each value have been included to the text's main body, according to the reviewer's suggestion |
Reviewer 3 Report
In general, the manuscript is of good quality, but it needs to be improved in many aspects before being published in “Sustainability”. Some comments below.
Line 19: Percentage of what? Important to cite what compounds this study is testing at the beginning of the abstract.
Throughout the manuscript. The concentrations must be presented in mass fraction (mg L-1 or µg L-1).
Lines 36-37: Specify more clearly what was statistically different.
Line 41: Caution, all compounds are organic pesticides.
Topic “Animal System: Mystus keletius”: The authors should organize the information more clearly. Information on some subjects is not in sequence, making the text confusing. Below are some examples.
Lines 147-149: Present this information with that provided on lines 138 and 139. In addition, please include references (Lines 138-139).
One information is not clear. "Fish were caught from ponds" (Lines 141-142) or "Fresh water catfish from a single population were obtained from a local pond" (Lines 150-151).
Line 151: (body length 15 1 cm, body weight 45 5 g). Dots must be included.
Line 153: mgL1 is not correct. The correct form is mg L-1
Topic “Acute toxicity”: Please include the references used to conduct the experiment.
Line 164: What was the criterion used to choose the concentrations? Describe in the text.
Line 164: Avoid using ppb. Authors must include the formal form: µg L-1
Line 192: Which concentration?
Line 211: mg/ml. It is important to standardize the presentation of concentrations throughout the manuscript. Previously it was displayed as mg L-1 and now mg/ml.
Topic “Glyceraldehyde dehydrogenase (GDH)”: Please include reference.
Table and Figs. Authors should avoid submitting duplicate results. Table 1 - Figs. 1 and Table 2 - Figs. 2-3-4 seem to show the same results. Authors must choose how they want to present the results only once; the other form of presentation must enter as supplementary material.
Table 1 and 2: The inclusion of abbreviations in full, at least in the caption, facilitates the reader's understanding.
Figures: Include the acronyms in full for all figures, at least in the captions, following the example of Figs. 1.c-d.
Table 1: Are all data significant at the p<0.05 level relative to controls?
Line 615: What represent (106)?
Author Response
In general, the manuscript is of good quality, but it needs to be improved in many aspects before being published in “Sustainability”. Some comments below.
|
Comment 1 |
: |
Line 19: Percentage of what? Important to cite what compounds this study is testing at the beginning of the abstract. |
|
Response |
: |
Effective concentration is expressed as a percentage. The abstract now contained a list of the chemicals utilised in the investigation |
|
|
|
|
|
Comment 2 |
: |
Throughout the manuscript. The concentrations must be presented in mass fraction (mg L-1 or µg L-1). |
|
Response |
: |
All concentrations throughout the documents are now presented in mass fractions |
|
|
|
|
|
Comment 3 |
: |
Lines 36-37: Specify more clearly what was statistically different. |
|
Response |
: |
All the results obtained were statistically significant (P<0.001) |
|
|
|
|
|
Comment 4 |
: |
Line 41: Caution, all compounds are organic pesticides. |
|
Response |
: |
Not all the compounds are organic pesticides. Ekalux and Impala are Synthetic (Chemical pesticide). Neemstar is an organic pesticide |
|
|
|
|
|
Comment 5 |
: |
Topic “Animal System: Mystus keletius”: The authors should organize the information more clearly. Information on some subjects is not in sequence, making the text confusing. Below are some examples. |
|
|
Lines 147-149: Present this information with that provided on lines 138 and 139. In addition, please include references (Lines 138-139). |
|
|
Response |
: |
The toxicity study used Mystus keletius (Catfish) because to its local availability, market demand, and good survival rate under laboratory settings. Reference No. 35 (Verma AK, Prakash S. Impact of arsenic on haematology, condition factor, hepatosomatic and gastrosomatic index of a fresh water cat fish, Mystus vittatus. International Journal on Biological Sciences. 2019;10(2):49-54) is added to support the statement |
|
|
|
|
|
Comment 6 |
: |
One information is not clear. "Fish were caught from ponds" (Lines 141-142) or "Fresh water catfish from a single population were obtained from a local pond" (Lines 150-151). |
|
Response |
: |
The authors apologise for the sentence's repetition. The repeated statement has now been deleted |
|
|
|
|
|
Comment 7 |
: |
Line 151: (body length 15 1 cm, body weight 45 5 g). Dots must be included. |
|
Response |
: |
The authors apologise for the sentence's repetition. The repeated statement contains these details, which has now been deleted |
|
|
|
|
|
Comment 8 |
: |
Line 153: mgL1 is not correct. The correct form is mg L-1 |
|
Response |
: |
The entire manuscript is proofread, and where necessary, the right form is updated |
|
|
|
|
|
Comment 9 |
: |
Topic “Acute toxicity”: Please include the references used to conduct the experiment. |
|
Response |
: |
Reference Number 36 (Glusczak L, dos Santos Miron, D, Moraes BS, Simões, RR, Schetinger MRC, Morsch VM, Loro VL (2007) Acute effects of glyphosate herbicide on metabolic and enzymatic parameters of silver catfish (Rhamdia quelen) Comparative Biochemistry and Physiology Part C: Toxicology Pharmacology 146(4):519-524) is added which is used to conduct the acute toxicity experiment |
|
|
|
|
|
Comment 10 |
: |
Line 164: What was the criterion used to choose the concentrations? Describe in the text. |
|
Response |
: |
Based on the EC50 values, we have fixed the concentration |
|
|
|
|
|
Comment 11 |
: |
Line 164: Avoid using ppb. Authors must include the formal form: µg L-1 |
|
Response |
: |
Throughout the manuscript, modifications are made in response to the reviewer's suggestions |
|
|
|
|
|
Comment 12 |
: |
Line 192: Which concentration? |
|
Response |
: |
Sub lethal concentration : Ekalux (EC-25%), Impala (EC-55%), NeemStar (EC-15%) |
|
|
|
|
|
Comment 13 |
: |
Line 211: mg/ml. It is important to standardize the presentation of concentrations throughout the manuscript. Previously it was displayed as mg L-1 and now mg/ml. |
|
Response |
: |
The entire manuscript has been reviewed for consistency in the presentation of concentrations, which have been adjusted to mgL-1 |
|
|
|
|
|
Comment 14 |
: |
Topic “Glyceraldehyde dehydrogenase (GDH)”: Please include reference. |
|
Response |
: |
Reference Number 46 (Sahib IKA, Rao KS, Rao KR (1983) Dehydrogenase systems of the teleost, Tilapia mossambica under augmented sublethal malathion stress Journal of Animal Morphology and Physiology 30(1):101-106) is added which is used to estimate Glyceraldehyde dehydrogenase (GDH) |
|
|
|
|
|
Comment 15 |
: |
Table and Figs. Authors should avoid submitting duplicate results. Table 1 - Figs. 1 and Table 2 - Figs. 2-3-4 seem to show the same results. Authors must choose how they want to present the results only once; the other form of presentation must enter as supplementary material. |
|
Response |
: |
Tables 1 and 2 have been relegated to the supplementary materials area as Tables 1S and 2S, respectively |
|
|
|
|
|
Comment 16 |
: |
Table 1 and 2: The inclusion of abbreviations in full, at least in the caption, facilitates the reader's understanding. |
|
Response |
: |
Changes have been made in response to the reviewer's comments, and a separate section on abbreviations has been included at the end of the manuscript |
|
|
|
|
|
Comment 17 |
: |
Figures: Include the acronyms in full for all figures, at least in the captions, following the example of Figs. 1.c-d. |
|
Response |
: |
Changes suggested by the reviewer has been carried out |
|
|
|
|
|
Comment 18 |
: |
Table 1: Are all data significant at the p<0.05 level relative to controls? |
|
Response |
: |
Yes, the data are all statistically significant when compared to the Control experiment. |
|
|
|
|
|
Comment 19 |
: |
Line 615: What represent (106)? |
|
Response |
: |
We apologise; this was a typographical error that has been corrected. |
|
|
|
|
Round 2
Reviewer 1 Report
The revision of the manuscript based to the first proof reading is not acceptable. The study focuses in specific compounds and examined mixtures of pesticides (commercial products) in a specific area. The article has serious flaws, additional experiments are needed and research is not conducted correctly or not explained in detail.
Authors should reconsider the methods used for this study and revise their manuscript accordingly.
Author Response
Reviewer #1:
|
Comment 1 |
: |
The revision of the manuscript based to the first proof reading is not acceptable. The study focuses in specific compounds and examined mixtures of pesticides (commercial products) in a specific area. The article has serious flaws, additional experiments are needed and research is not conducted correctly or not explained in detail. Authors should reconsider the methods used for this study and revise their manuscript accordingly.
|
|
Response |
: |
We appreciate your insightful remarks. As stated in the abstract and throughout the text, the study was undertaken to determine the hazardous effect of three pesticides on freshwater catfish Mystus keletius, namely Ekalux (EC-25 %), Impala (EC-55 %), and NeemStar (EC-15 %).
The experimental section was conducted using previously published and acceptable procedures, and so we do not believe that a re-evaluation of the methodology for this work is essential. |
|
|
|
|
Reviewer 2 Report
The authors have made progress on the manuscript and manuscript is very interesting. However, the presentation needs to be improved. For example, the below paragraph talks about hemoglobin concentration but there are numerous grammar issues and no units (there must be units with the values) in this one paragraph (and throughout the entire manuscript). This was my initial concern with the manuscript and is still my main concern as this has not been addressed by this revision.
Hb content in the experimental fishes exposed to Ekalux on the day 7 recorded the value of 6.59 that reached the final value of 2.00 on the day 28 indicating the declining trend. A similar trend was observed for Impala and Neem star where the value in the treated fishes was 4.03 that reach a value of 1.65 on the day day 28 in follower and 6.73 on the day 7 priest minimum value of 2.75 on the day 28 in response to the pesticides Neemstar (Table 1S; Fig.1c).
Some other examples:
Line 145-147: Awkward as written. Please consider the following:
Mystus keletius (Catfish) was chosen for this study due to its increasing prevalence and usefulness as a subject for toxicology testing. This fish species also demonstrates an excellent ability to adapt to changing environmental conditions, as well as tolerate laboratory conditions [35].
Line 149-150: Awkward as written. Please consider the following:
All Mystus keletius fingerlings used in this study weighted between 3.6–10.0g and measured between 4.0–8.0cm in total length.
Line 155-156: These two sentences convey the same meaning as lines 145-147. May want to consider removing.
Line 195: Spell out first work in sentence.
Mystus keletius (3.45 ± 0.74 g) were exposed to sub lethal concentration of pesticides
Line 203: If possible, please describe more precisely what “properly cleaned” means (what are the steps). This should be added if cleaning the fish at this step could alter downstream results. If not, then this is ok.
Line 245-248: Awkward as written. Please consider the following:
The findings in this article were reported as mean and standard error. The two-tailed student’s t-test was used to determine statistical significance (p<0.05) between the control and experimental groups.
Line 251-253: Awkward as written. Please consider the following and change throughout the manuscript and add units as the values don't convey meaning without the units.
On RBC, a similar trend was observed for Impala, which started at 1.36 on day 7 and reached a value of 1.10 on day 28, while Neemstar's value decreased from 252 1.48 on day 7 to 1.02 on day 28 (Fig. 1a).
Author Response
Comments of Reviewer #2:
The authors have made progress on the manuscript and manuscript is very interesting. However, the presentation needs to be improved. For example, the below paragraph talks about hemoglobin concentration but there are numerous grammar issues and no units (there must be units with the values) in this one paragraph (and throughout the entire manuscript). This was my initial concern with the manuscript and is still my main concern as this has not been addressed by this revision.
|
Comment 1 |
: |
Hb content in the experimental fishes exposed to Ekalux on the day 7 recorded the value of 6.59 that reached the final value of 2.00 on the day 28 indicating the declining trend. A similar trend was observed for Impala and Neem star where the value in the treated fishes was 4.03 that reach a value of 1.65 on the day day 28 in follower and 6.73 on the day 7 priest minimum value of 2.75 on the day 28 in response to the pesticides Neemstar (Table 1S; Fig.1c). |
|
Response |
: |
Kindly accept our apologies for failing to include your comments during the revising process. It is not deliberate, but an error. We have updated the entire text to reflect the new manuscript's units and values |
|
|
|
|
|
Comment 2 |
: |
Line 145-147: Awkward as written. Please consider the following: Mystus keletius (Catfish) was chosen for this study due to its increasing prevalence and usefulness as a subject for toxicology testing. This fish species also demonstrates an excellent ability to adapt to changing environmental conditions, as well as tolerate laboratory conditions [35]. |
|
Response |
: |
Changes were made in response to the reviewer's suggestion |
|
|
|
|
|
Comment 3 |
: |
Line 149-150: Awkward as written. Please consider the following: All Mystus keletius fingerlings used in this study weighted between 3.6–10.0g and measured between 4.0–8.0cm in total length. |
|
Response |
: |
Changes were made in response to the reviewer's suggestion |
|
|
|
|
|
Comment 4 |
: |
Line 155-156: These two sentences convey the same meaning as lines 145-147. May want to consider removing. |
|
Response |
: |
Changes were made in response to the reviewer's suggestion |
|
|
|
|
|
Comment 5 |
: |
Line 195: Spell out first work in sentence. Mystus keletius (3.45 ± 0.74 g) were exposed to sub lethal concentration of pesticides |
|
Response |
: |
Changes were made in response to the reviewer's suggestion |
|
|
|
|
|
Comment 6 |
: |
Line 203: If possible, please describe more precisely what “properly cleaned” means (what are the steps). This should be added if cleaning the fish at this step could alter downstream results. If not, then this is ok. |
|
Response |
: |
Changes were made in response to the reviewer's suggestion |
|
|
|
|
|
Comment 7 |
: |
Line 245-248: Awkward as written. Please consider the following: The findings in this article were reported as mean and standard error. The two-tailed student’s t-test was used to determine statistical significance (p<0.05) between the control and experimental groups. |
|
Response |
: |
Changes were made in response to the reviewer's suggestion |
|
|
|
|
|
Comment 7 |
: |
Line 251-253: Awkward as written. Please consider the following and change throughout the manuscript and add units as the values don't convey meaning without the units. On RBC, a similar trend was observed for Impala, which started at 1.36 on day 7 and reached a value of 1.10 on day 28, while Neemstar's value decreased from 252 1.48 on day 7 to 1.02 on day 28 (Fig. 1a). |
|
Response |
: |
Changes were made in response to the reviewer's suggestion |
|
|
|
|
Reviewer 3 Report
The presentation of concentrations must be standardized. In the current version there are different writing formats.
Line 127: g/mol
Line 142: g·mol−1
Line 167: μgL-1.
Line 254: g/dL
Abbreviations: mg.l-1
Line 41: All pesticides evaluated in this study are organic molecules. So avoid making synthetic vs. organic comparisons as this tends to confuse readers. The most appropriate comparison would be synthetic vs. natural.
Author Response
|
Comment 1 |
: |
The presentation of concentrations must be standardized. In the current version there are different writing formats. Line 127: g/mol Line 142: g·mol−1 Line 167: μgL-1. Line 254: g/dL Abbreviations: mg.l-1 |
|
Response |
: |
The concentration units are now corrected and consistent throughout the revised manuscript |
|
|
|
|
|
Comment 2 |
: |
Line 41: All pesticides evaluated in this study are organic molecules. So avoid making synthetic vs. organic comparisons as this tends to confuse readers. The most appropriate comparison would be synthetic vs. natural. |
|
Response |
: |
As suggested by the reviewer, the term organic has been substituted with the term natural throughout the revised manuscript |
|
|
|
|
Round 3
Reviewer 2 Report
The authors have made improvements; however, my first review comment still stands in that the article would be vastly improved by having it reviewed by a native English speaker or a language service. I enjoy reading this manuscript and think it has great value but please carefully review the manuscript for all inconsistencies not just the one's below as they are examples. The entire manuscript needs to be reviewed for inconsistencies.
Also, after reading this version, some of the confusion and awkward narrative in the results could be moved into one or two larger tables to provide values per time frame with the units. Perhaps just use the result's narrative to talk about trends and statistics. This would eliminate the need to repeat the units frequently.
Examples:
Lines 27-29ish: Consider revising to: ….on RBC content on day 7 was 1.43 million/mm3 and decreased to 1.18 million/mm3 on day 14 and 21 respectively. A similar trend was observed for Impala on RBC which had an initial value of 1.36 million/mm3 on day 7 and declined to 1.10 million/mm3 on day 28. However, in the case of Neemstar, the… (I am not sure what the authors are stating at this point).
Line 40: Sentence is awkward as written. Consider revising to something like: Overall, Impala had the strongest effect on the recorded cellular enzymes in this study followed by Ekalux and Neemstar.
Line 59: Consider removing the – out of biopesti-cides
Line 99: Consider revising to something like: The pesticides Ekalux, Impala and Neemstar were evaluated in this study and were purchased from local farm stores.
Line 148: This sentence is the same as line 146 so please remove. Suggests that the edits were rushed.
Line 151: Please changed weighted to weighed
Line 151 and Line 152: 3.6-10g and 4.0-8.0cm… Please revise to 3.6-10 g and 4.0-8.0 cm
Line 197: Consider revising to: …prepared pellet food once a day for…
Line 220, 221, 223: unit spacing is inconsistent.
Line 228 and Line 237: unit spacing is inconsistent.
Line 248-252: Which pesticide for the first sentence? It is unclear.
Line 250 and 252: Consider revising to: … on day 7
Line 252: Consider revising to: …on day 28
Line 256-258: Please add the units after the result values.
Line 261: Consider revising to: …on day 28
Line 264: Please revise this sentence as this does not make sense? …on the day day 28 in follower and 6.73 g/dL on the day 7 priest minimum value of 2.75 g/dL on the day 28…
Line 267: Please be consistent with the capitalization of Ekalux and please make all the phrase “on the day” to “on day” throughout the entire manuscript.
Figure 1: Please consider revising the chart titles as they are awkward as written.
Suggest something like:
Effect of selected pesticides on red blood cell levels in the blood of Mystus keletius.
Effects of selected pesticides on white blood cell levels in the blood of Mystus keletius
Ect…
Line 306, 310, 314: Consider revising to: On day 7…
Line 318-320: This sentence is awkward as written.
Line 331: Please be consistent with the capitalization of ekalux.
Line 335: Please use lower case for “gill”
Similar issues throughout the remaining portions of the manuscript.
Author Response
Thanks for your comments, please check the attachment.

Round 4
Reviewer 2 Report
Ecotoxicological effects of pesticides on haematological parameters and oxidative enzymes in freshwater catfish, Mystus keletius
The authors have made progress on the manuscript. The first half of the manuscript reads very well, however, starting from section 3.2 (Effect of Impala on Respiratory Enzymes) the readability declines with numerous grammar issues throughout the remaining part of the manuscript as if this part was not thoroughly reviewed/revised by the authors. As previously mentioned, please consult with a native English speaker and take time to do all the edits. As mentioned, the first part of the manuscript has improved a lot.
The introduction is a little short as well. Please add a third paragraph stating the goals of the experiment which is missing. The goals are in the discussion but should be in the introduction.
Examples where grammar makes if difficult to read. This is not an exhaustive list, please review the entire manuscript for grammar.
line 369-370: When analyse statistical e the data indicate that all the values where significant at p˂0.05.
Several places have “ statistical e” ? Here and in the Line 464-468 example below.
Line 336-41:
Similarly MDH activity also recorded a declining trend starting from 0.91 mg reduced TTC/g /wet wt/h to on day 7 to a minimum of 0.65 mg reduced TTC/g /wet wt/h on day 28 likewise the activity exhibited a variation where there was a slight improvement observed on day 7 that with continuous increase in activity and reach a maximum on day 28 indicating that the activity of GDH had an inverse relationship as a function of time in the brain tissue (Table 2S; Fig. 3a).
This should be broken up into multiple concise sentences, also check grammar.
Line 393-396:
Data indicates that there is declining trend was observed for SDH and MDH activities concerned and an inverse relationship was exhibited by GDH activity in brain tissues (Table 2S; Fig. 4a). When analysed for its activity SDH, MDH and GDH showed a differential value in gill issue.
Hard to read, grammar issues, and needs to be written in the past tense. The last sentence does not really convey any information, please consider better word choice.
Line 464-468:
Likewise MDH in muscle tissue had a value of 0.86 on the 7th that reach a minimum value of 0.4 mg reduced TTC/g /wet wt/h on the 28th day with intermediate values on 21st and 14th day respectively activity GDH activity e exhibited an increasing trend with a value of 1.43 on 7th day and it's a maximum of 2.15 on the 28th day with intermediate values on 14th day and a value of 2.05 mg reduced TTC/g /wet wt/h on 21st day.
Line 470-480:
Effect of Neem star on different tissue system over the treatment period in the experimental animal had interesting change in the activity pattern of enzymes from 0 to 28th day of treatment. on the 7th day is SDH activity recorded the value of 0.6 to that reached value of 0.5 to on the 28th day with 0.57 on the 14th day and 0.55 on the 21st day a similar trend was observed for MDH activity that had a minimum value of 0.57 on the 28th day with value corresponding to 0.74, 0.89 ,0.98 on 21st 14th and 7th respectively GDH activity showed an increasing trend starting from a value of 0.77 on the 7th day increased to 0.82 on 14th day 1.04 on 21st day and reached a maximum of 1.19 on the 28th the day. Data indicates that there is declining trend was observed for SDH and MDH activities concerned and an inverse relationship was exhibited by 479 GDH activity in brain tissues.